# Lipid droplet distribution quantification method based on lipid droplet detection by constrained reinforcement learning

Yoshitomi Harada[1], Haruto Nishida[2]*, Keiko Matsuura[3]

**1** Faculty of Health and Medical Sciences, Nippon Bunri University, Oita, Japan, **2** Diagnostic Pathology, Faculty of Medicine, Oita University, Oita, Japan, **3** Department of Biomedicine, Faculty of Medicine, Oita University, Oita, Japan

* nharuto@oita-u.ac.jp

## Abstract

We previously proposed the lipid droplet detection by reinforcement learning (LiDRL) method using a limited dataset of pathological images. The method automatically detects lipid droplets using reinforcement learning to optimize filter combinations based on their size and grayscale contrast. In this study, we aimed to detect lipid droplets reliably and analyze their distribution patterns across pathological tissue images. For this purpose, we improved the environmental and agent-side functions in LiDRL to obtain a revised method. These improvements increased the stability and robustness of the system, enabling consistent extraction of lipid droplets of similar sizes across all rank levels in the pathological tissue images. We quantified the lipid droplet distribution using average probability density and entropy and visualized it as a heat map. This analysis facilitates the extraction of lipid droplet characteristics that could serve as indicators of liver disease.

## Introduction

Lipid droplets are intracytoplasmic structures that store neutral lipids and play important roles in cellular energy storage and metabolism [1]. Lipid dysregulation is associated with various diseases, including obesity, diabetes, and fatty liver. Therefore, researchers have proposed novel methods to quantify lipid droplets, which are involved in metabolic dysfunction-associated fatty liver disease [2–4].

In pathological tissue images, lipid droplets exhibit complex features—including variations in size, shade, and shape—making rapid and accurate assessment challenging even for experienced pathologists. To address this challenge, several neural network-based methods have been proposed [5–14]. Advanced techniques, such as convolutional neural networks, are suitable for image classification but struggle to preserve fine structures in segmentation tasks [15–17]. U-Net-based methods have been proposed; however, they come with inherent risks such as increased

**Data availability statement:** The data are available at Zenodo (https://doi.org/10.5281/zenodo.15733729).

**Funding:** The author(s) received no specific funding for this work.

**Competing interests:** The authors have declared that no competing interests exist.

computational complexity and memory requirements, and overfitting due to the symmetric structure of the encoder and decoder [18–20]. Consequently, these methods were deemed unsuitable for this study and therefore were not utilized. Moreover, the utilization of advanced imaging techniques necessitates substantial time and resources to analyze numerous images after extensive model training [5–20]. This aspect can be particularly challenging when examining rare diseases or lesions. Therefore, to automatically detect targets with limited data, we developed the lipid droplet detection by reinforcement learning (LiDRL) method, which employs reinforcement learning to extract candidate lipid droplets, based on a classifier trained on simplified features [21].

However, the LiDRL method encounters limitations in certain hierarchical layers, where lipid droplets cannot be reliably separated based on size. To reliably extract lipid droplets from each hierarchical layer, we modified the environment and enhanced the functionality of the reinforcement learning agent. Subsequently, we quantified and visualized the distribution of the lipid droplets by performing average probability density and entropy analyses using kernel density estimation (KDE) based on the center of gravity of the lipid droplets detected with the revised method.

## Materials and methods

### Ethics approval and patient consent

All procedures were performed in compliance with relevant laws and institutional guidelines. Experimental images were acquired adhering to ethical research standards (Oita University Faculty of Medicine Ethics approval no. 2568/15 June 2023). Informed consent was waived by the Medical Ethics Review Board. The study was retrospective, and patient consent was obtained using an opt-out method. Specimens were obtained from the archives of the institute, with no additional invasive procedures performed. The privacy rights of human subjects were protected throughout the study. The study was conducted from June 15, 2023, to March 31, 2025.

### Materials

Hematoxylin–eosin specimens with lipid droplets, anonymized and prepared at the Department of Diagnostic Pathology, were photographed using an optical microscope (OLYMPUS BX51-TF, 20×, 40×), an objective lens (UPlanSApo), and an OLYMPUS DP73 camera. These specimens were obtained from surgical non-tumoral specimens, excluding those for hepatocellular carcinoma. The study involved three patients, with two images captured from each, totaling to six images. These images featured the central and portal veins (8-bit grayscale images with a resolution of 1600×1200 pixels). The images were randomly selected based on anatomical requirements (one portal vein region, one central vein region). However, the results of an analysis using a limited number of images may have limited applicability when extrapolated to the entire patient population. Therefore, we estimated the shape of the distribution using KDE plots and quantified the randomness and information content of the dataset using entropy analysis. KDE enabled effective estimation of

data distribution patterns despite the small sample size, facilitating more accurate feature extraction. Entropy analysis, which evaluates data complexity—including diversity and bias—provided critical insights for appropriate model selection in subsequent analysis.

To shorten the processing time, the number of pixels in both the vertical and horizontal directions was reduced by half, and the gradation was compressed to 8 bits, with a brightness value of 0 representing black and 255 representing white. The minimum area of lipid droplets for detection was set at 40 pixels, and the aspect ratio of the detection area was set at 2.0. Areas with an aspect ratio smaller than 1.5 were considered non-lipid droplets and excluded from the study. Python 3.10.12 was used for analysis and heat mapping. We use the following terms to explain our algorithm: filter size, a square frame with one side as large as the maximum diameter that can contain only one lipid droplet; rank, order within a 3 × 3 filter region in a binary image; environment, a 4 × 10 grid; reward, the ratio of overlapping region to detected lipid droplet region (Fig 1); agent, a subject that moves, learns, and selects actions that maximize rewards based on the state of the environment; state, the agent's position in the environment, defined in terms of filter size and rank; action, the possible moves the agent can take from each state, i.e., the agent can move leftward or downward; Q-value, the total reward expected from a specific action in a given state; penalty area, an area that, if entered, leads to a negative reward, with the agent forced to return to the starting point. This area maintains accuracy by repeating a certain number of movements; start, the initial position of the agent; goal, the final position of the agent.

## Conventional lipid droplet detection method

The LiDRL method simplifies the size (filter size) and grayscale contrast (rank) of the lipid droplets in tissue images as lipid droplet information and employs Q-learning using a learning agent. A 4 × 10 grid was used as the environment, and the lipid droplets were detected by combining the behavior of the agent with the environment. A fixed penalty area representing a negative reward was placed within this environment (Fig 2). In this environment, the agent (red circle in Fig 2) interacts with multiple penalty areas (indices 7, 14, and 33 in Fig 2 and the position of the red area), navigating from the start position in the upper right corner to the goal position in the lower left corner. By finding a path that maximizes the reward, the agent distinguishes lipid droplets of similar size by categorizing them into four hierarchical levels (Ranks 6, 5, 4, and 3). The overlap between the detected lipid droplets was designated as the Q-value for the reward.

## Detection and distribution methods

**Experiment 1: Improvement of the LiDRL method (revised LiDRL method).** We improved the LiDRL method through agent behavior learning using constraints. First, we aimed to reliably extract lipid droplets of similar size and grayscale contrast from images. To ensure the accurate extraction of lipid droplets from each rank, we implemented measures to prevent continuous downward movement.

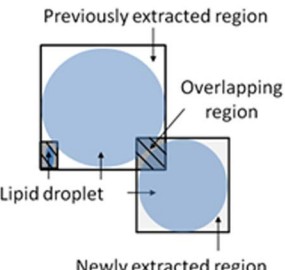

**Fig 1. Definition of reward: Overlapping region of the newly extracted region [21].**

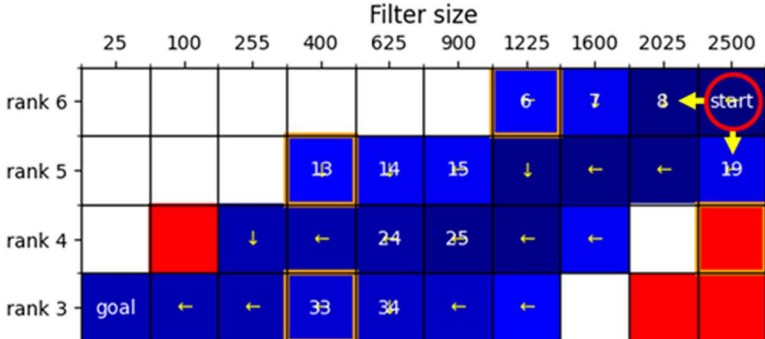

**Fig 2. Learning environment for agents.** An example of the learning environment for agents in the conventional lipid droplet detection using reinforcement learning (LiDRL) and revised LiDRL methods is presented. The numbers in each coordinate represent the index, and the arrows represent the agent's behavioral direction after learning. The density of blue in each index is proportional to the natural logarithm of the number of times the agent learns. White represents zero learning times, red represents the fixed penalty area, and orange represents the final position of the moving penalty area.

In this revised method, penalty areas were initially placed at indices 8, 15, and 34 within the LiDRL search range, allowing the agent to begin learning within a narrow range. These penalty areas move to the left based on the number of collisions with the agent, denoted as thresholds s1, s2, and s3. The range of movement is limited: the areas at indices 8 and 15 can move up to two positions to the left, whereas the area at index 34 can move only one position to the left. However, the area at index 19 moves downward when the collisions with the agent exceed a threshold t. After a collision, the penalty area relocates to a different index. This enables the next agent to move to a penalty area that was initially inaccessible.

**Experiment 2: Analysis of lipid droplet distribution (LiDRL Q-method).** After Experiment 1, we analyzed the lipid droplet distribution and performed probability density distribution analysis and entropy analysis using KDE. We extracted lipid droplets using the revised LiDRL method, capturing the distribution of lipid droplet centroids at each rank using KDE [22]. KDE was used to capture the distribution of lipid droplets as clusters (Equation 1):

$$\hat{\lambda}(u) = \frac{1}{q(u)} \sum_{i=1}^{N} \frac{1}{h^2} \kappa \left( \frac{x_i - u}{h} \right)$$

(1)

where $\hat{\lambda}(u)$ denotes the estimate of point density at an arbitrary point u, $\kappa \left( \frac{(x_i-u)}{h} \right)$ the kernel function, h the Bandwidth, q(u) the Boundary correction, and N the number of lipid droplets. The kernel function used was a Gaussian function, and the bandwidth was fixed at 0.5.

The uncertainty and randomness of the data were quantified using entropy H (X) in Equation 2, derived from the probability density distribution of lipid droplets obtained through KDE [23].

$$H(X) = -\sum_i p(x_i) \log_e p(x_i)$$

(2)

where $p(x_i)$ represents the probability density at the centroid $x_i$ of each lipid droplet.

The probability density distribution was visualized using a heat map. The darker the color, the higher the density. Two pathologists independently reviewed the distribution of lipid droplets in the tissue images and evaluated the heat map results. The average values obtained from the six images were evaluated using the Mann–Whitney U test, p-value, and standard error to determine whether there were significant differences between the anatomical regions.

**Experiment 3: Comparison between the conventional and revised LiDRL methods.** We compared the performance of the conventional and revised LiDRL methods, adapting experimental and computational protocols accordingly. The performance of both models was evaluated in terms of sensitivity, specificity, accuracy, precision, recall, F1 score, and cross-validation (CV) F1 score (Equations 3 to 9), obtained over an average of five runs. In addition, as the performance of the conventional LiDRL method [21] was evaluated using the F1 score based on 10-fold CV, the same comparison criteria were used for the revised LiDRL method.

$$\text{Sensitivity} = \text{TP} / (\text{TP} + \text{FN}) \tag{3}$$

$$\text{Specificity} = \text{TN} / (\text{TN} + \text{FP}) \tag{4}$$

$$\text{Accuracy} = (\text{TP} + \text{TN}) / (\text{TP} + \text{TN} + \text{FP} + \text{FN}) \tag{5}$$

These measures represent the overall correctness of the classification model.

$$\text{Precision} = \text{TP} / (\text{TP} + \text{FP}) \tag{6}$$

It evaluates the proportion of predicted positives that are truly positive.

$$\text{Recall} = \text{TP} / (\text{TP} + \text{FN}) \tag{7}$$

It indicates the ability of the model to accurately capture true positives.

$$\text{F1 score} = 2 \times \text{Precision} \times \text{Recall} / (\text{Precision} + \text{Recall}) \tag{8}$$

It balances precision and recall, which is useful when classes are imbalanced.

$$\text{Cross} - \text{validation F1} - \text{score} = \frac{1}{k}\sum\nolimits_{i=1}^{k} \text{F1}_i (k = 10) \tag{9}$$

TP (True Positive): Correctly predicted positives
TN (True Negative): Correctly predicted negatives
FP (False Positive): Incorrectly predicted positives
FN (False Negative): Incorrectly predicted negatives

Furthermore, to compare and evaluate the performance of the revised method against the conventional method, two pathologists identified lipid droplets on the same images.

## Results

### Experiment 1

We tracked the agent's learning process based on changes in the Q-value, specifically within the penalty areas where collisions occurred. The collision number (threshold) for each combination of agent and penalty area is summarized in Tables 1 and 2 and Fig 3. The conventional LiDRL method failed to detect lipid droplets, resulting in no hierarchical structure (no lipid droplet layers) in some images (Table 1: Rank 5).

The average number of learning iterations and standard deviation were found to be 3181±233.8 and 3258±162.1, respectively, after running the conventional and revised LiDRL methods 100 times on the same image (Tables 1 and 2).

**Table 1. Comparison of lipid droplet detection counts in the same image [Image003].**

| Hierarchy | Conventional LiDRL method | Revised LiDRL method |
| --- | --- | --- |
| Rank 6 | 23 | 26 |
| Rank 5 | 0 | 20 |
| Rank 4 | 257 | 239 |
| Rank 3 | 205 | 207 |

**Table 2. Comparison of the number of actions.**

| | Conventional LiDRL method | Revised LiDRL method |
| --- | --- | --- |
| mean±standard deviation | 3181±233.8 | 3258±162.1 |

A lipid droplet rank was not observed 19 times using the conventional method, but it was observed using the revised method. The agent may fall into a local solution while passing through the hierarchy because of learning, despite actively searching for lipid droplets at each rank. Therefore, to train the agent effectively, we tracked its behavior for each index (Fig 3). We found that the Q-value eventually fluctuated at each index, and the action with the highest Q-value was selected as the agent's action.

The blue (left) and orange (down) lines at each index represent the changes in the Q-value. This indicates the movement of the agent to the left or downward. At each index, the Q-value alternates, and the action with the highest Q-value is selected as the action of the agent (episode 350).

When each threshold is equal, the cumulative reward dropped once between approximately 75 and 140 learning iterations (indicated by the red arrow), followed by recovery and increase (Fig 4). Indices 24 and 25 in Fig 2 mark stages during which the agent learns to avoid penalty areas (shown at indices 33 and 34) along its path to the goal. Therefore, the agent learned situational rules, such as avoiding penalty areas, by combining simple leftward and downward actions in a complex manner.

## Experiment 2

The revied LiDRL method was used to calculate the average probability density values using the KDE from the lipid droplet distribution images by rank in the central and portal vein regions (Table 3). Lipid droplets in Ranks 6 and 5 were relatively low compared with those in Ranks 4 and 3, but the average probability density was high. Conversely, Ranks 4 and 3 contained more droplets but had lower average probability densities. This indicates that a small number of large lipid droplets were concentrated within a certain range. Conversely, the high number of lipid droplets and low average probability density indicate that medium-to-small droplets are more widely dispersed.

The average entropy was calculated for each tissue image used in Experiment 1 (Table 3). The entropy of the lipid droplet distribution was lower for Ranks 6 and 5 compared with that for Ranks 4 and 3, indicating that the distribution of the large lipid droplets was less disordered. Conversely, the distribution of medium-to-small lipid droplets observed in Ranks 4 and 3 was irregular (randomly distributed) in the tissue image. Therefore, the KDE-derived average probability density and entropy of the probability density distribution depend on the concentration and scattering of lipid droplets and the disorder and uncertainty in the lipid droplet distribution, respectively. A more comprehensive understanding of lipid droplet distribution characteristics can be achieved by applying and jointly evaluating both methods from different analytical perspectives. This enabled the quantification and visualization of lipid droplet distribution via a heat map. Representative examples of the distribution detected by the revised LiDRL method are shown in Fig 5 (central vein region) and Fig 6 (portal vein region).

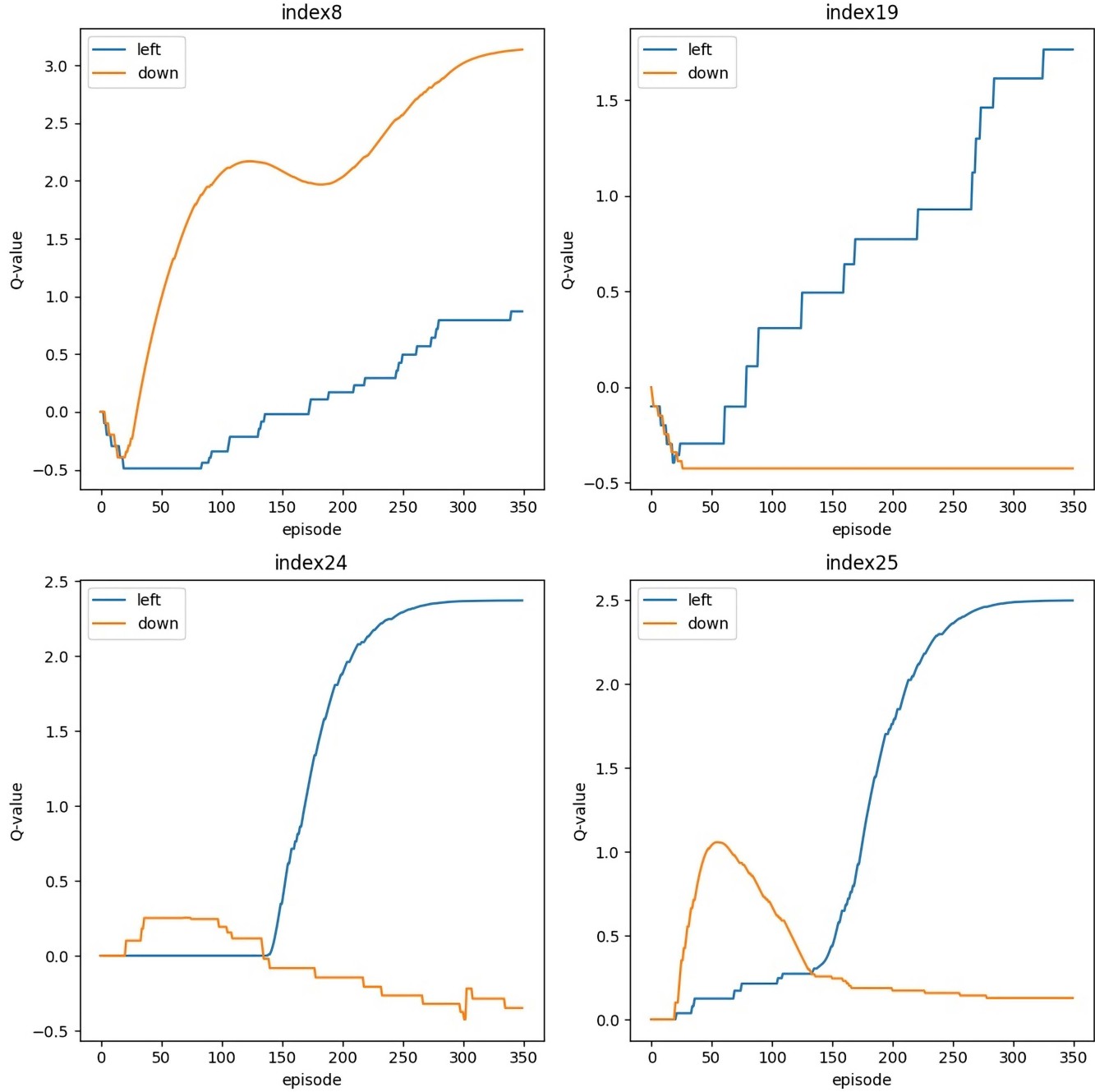

**Fig 3. Changes in Q-value at each index (s1 = 10, s2 = 10, s3 = 10, t = 10).**

## Experiment 3

Compared with lower ranks, higher ranks tended to exhibit a distribution pattern with fewer lipid droplets, higher average probability density, and lower entropy levels (Table 3). The average probability density was lower in the central vein region than in the portal vein region, while entropy was higher. Finally, two pathologists verified the morphological findings,

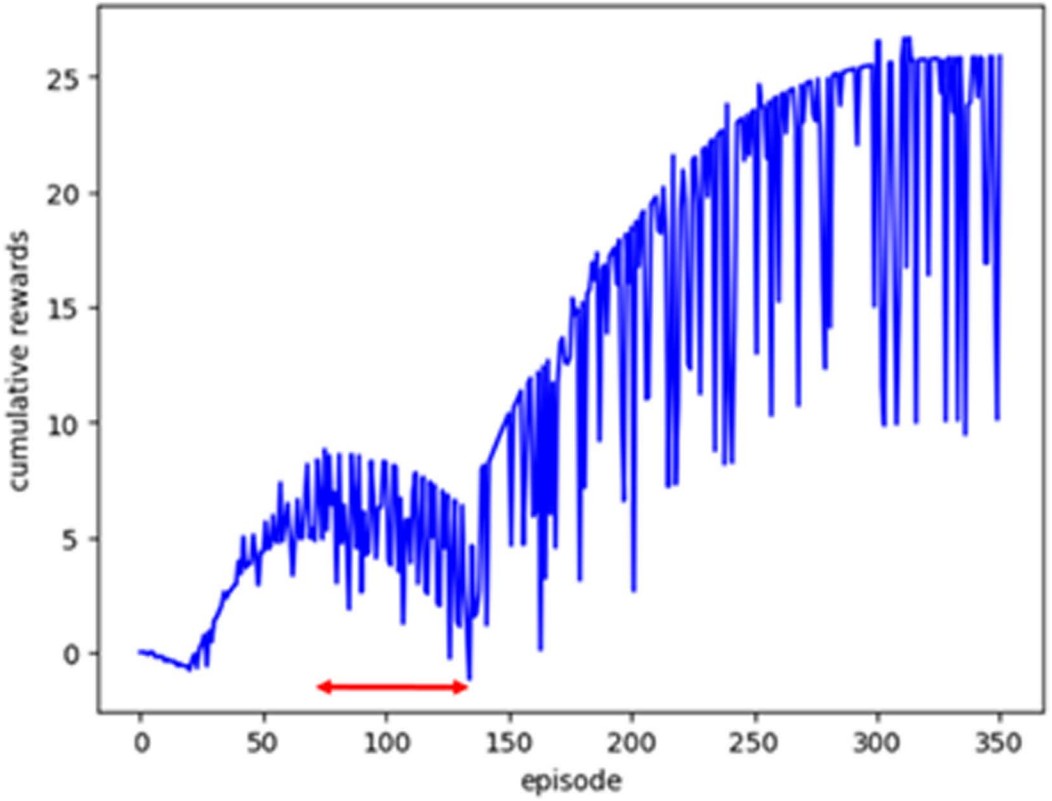

**Fig 4. Accumulative reward (s1 = 10, s2 = 10, s3 = 10, t = 10).** The red arrow indicates the area where the agent modifies its search for a route, leading to a decrease in the cumulative reward for the Q-value.

**Table 3. Variations between the central vein region and the portal vein region [six images].**

| Category | Hierarchy | Central vein region | Portal vein region | Mann-Whitney U | p-value | Relative Error [%] |
|---|---|---|---|---|---|---|
| **Lipid Droplets Count** | Rank 6 | 36.0 | 23.3 | 7.0000 | 0.4000 | 35.19 |
| | Rank 5 | 55.7 | 12.7 | 6.0000 | 0.5050 | 6.59 |
| | Rank 4 | 241.0 | 256.0 | 4.0000 | 1.0000 | 6.22 |
| | Rank 3 | 283.3 | 220.3 | 8.0000 | 0.2000 | 22.24 |
| **Average Probability Density** | Rank 6 | 0.0186 | 0.0296 | 2.0000 | 0.4000 | 59.29 |
| | Rank 5 | 0.0115 | 0.0637 | 0.0000 | 0.0765 | 452.13 |
| | Rank 4 | 0.0027 | 0.0026 | 5.0000 | 1.0000 | 3.80 |
| | Rank 3 | 0.0023 | 0.0030 | 1.0000 | 0.2000 | 33.57 |
| **Average Entropy** | Rank 6 | 2.557 | 2.268 | 7.0000 | 0.4000 | 11.29 |
| | Rank 5 | 2.844 | 1.831 | 9.0000 | 0.0765 | 35.62 |
| | Rank 4 | 3.777 | 3.808 | 4.0000 | 1.0000 | 0.84 |
| | Rank 3 | 3.881 | 3.709 | 8.0000 | 0.2000 | 4.44 |

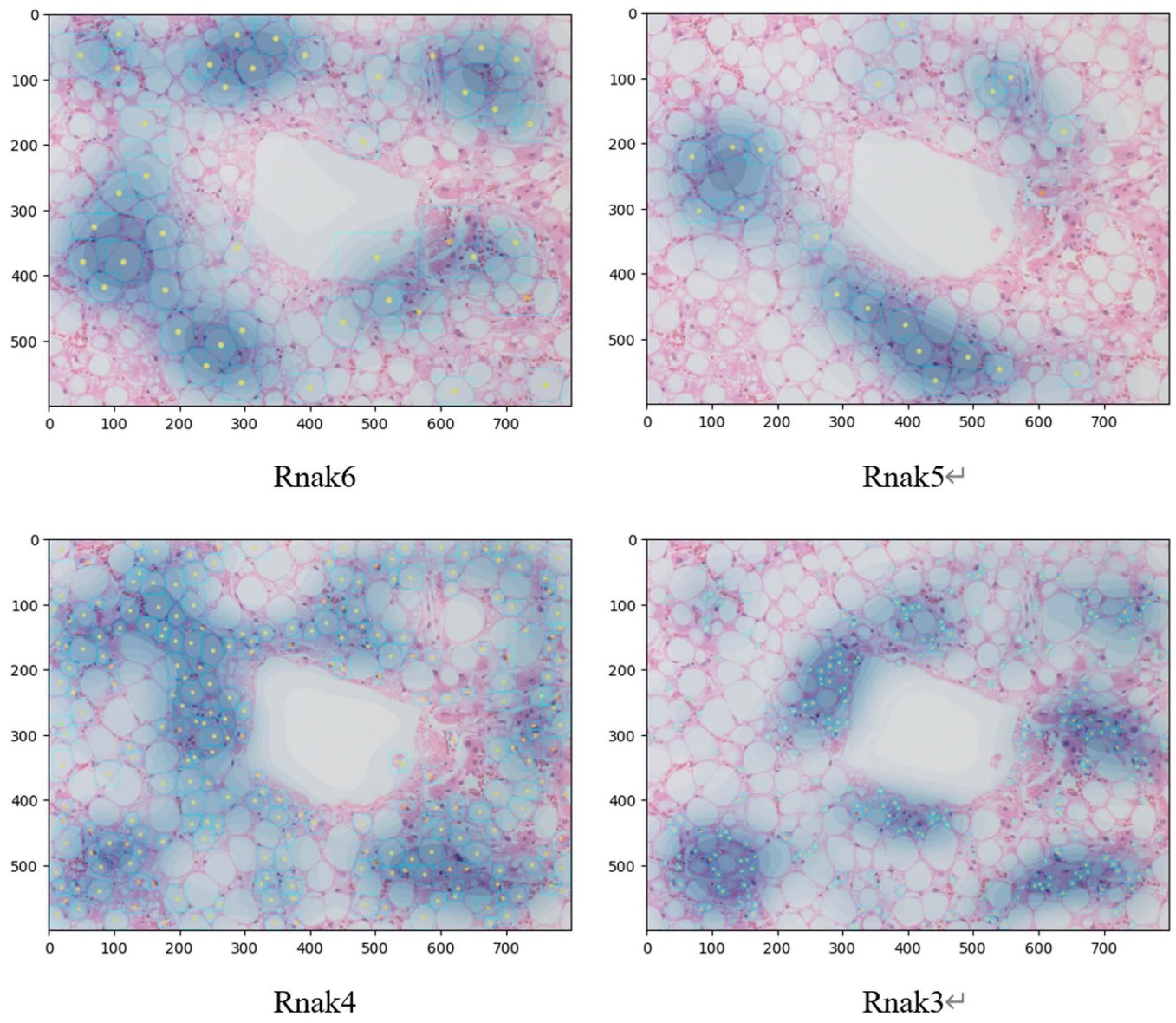

**Fig 5. Example heat map image of revised lipid droplet detection using reinforcement learning (central vein region).** For each rank, the blue squares indicate the detected lipid droplets, yellow dots indicate the centers of gravity of the detected lipid droplets, and blue shading indicates the lipid droplet distribution evaluated by kernel density estimation.

and the analysis results of the lipid droplet distribution in both cases were generally consistent with the morphological assessment.

## Discussion

The conventional LiDRL method did not detect lipid droplets in certain hierarchies within the images. This issue could be attributed to the agent exhibiting a specific behavior (local solution), which facilitated achieving a higher reward as the rank decreased owing to the extraction of finer images, such as noise. The agent might have passed through the

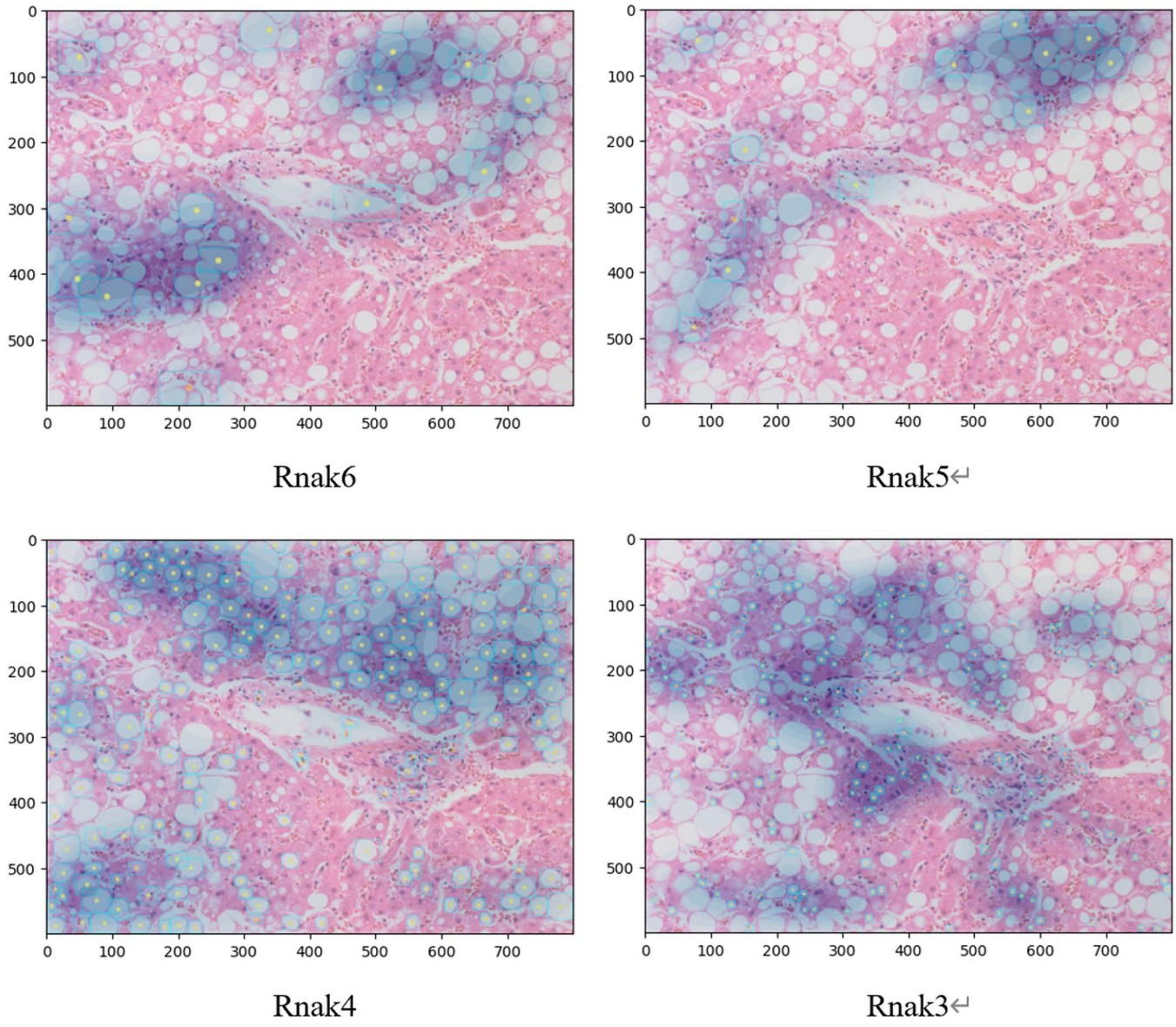

**Fig 6. Example heat map image of revised lipid droplet detection using reinforcement learning (portal vein region).** For each rank, the blue squares indicate the detected lipid droplets, yellow dots indicate the centers of gravity of the detected lipid droplets, and blue shading indicates the lipid droplet distribution evaluated by kernel density estimation.

hierarchy and reached a local solution because the accuracy of lipid droplet extraction increased as the rank corresponding to the threshold for binarization decreased. In the early stages of learning, the movement towards the moving penalty area was dominant. The predominant movement to the left from the starting position can likely cause blank areas, which refers to the penalty effect when the agent moves downward twice consecutively. Therefore, an environment with designated movable penalty areas was created.

A decrease in cumulative reward during early learning stages was similarly observed in the conventional LiDRL method. The revised method exhibited a smaller standard deviation in terms of the number of learning cycles than that in the conventional method, indicating more stable learning. Furthermore, the revised LiDRL method changed the initial

conditions and settings, performed learning under strict conditions, and increased the number of iterations. No significant change was observed in the number of lipid droplets detected, facilitating the learning of various patterns and features, even under diverse and unknown conditions, such as new images. After the learning process, we eliminated actions from the Q-table that involved continuous downward movement through the ranks. It was evident that the agent learned by repeating actions to determine alternative routes to adapt to the environment (Experiment 1).

To capture the hierarchical distribution of the lipid droplets using the revised LiDRL method, the center of gravity of each lipid droplet was represented as a probability density using KDE (Experiment 2). The revised method uses a Gaussian function as the kernel, which can support distant observation points. The average probability density obtained using KDE was higher for Ranks 6 and 5, which had a smaller number of lipid droplets than for Ranks 4 and 3. This indicates that the average probability density reflects a distribution method based on the size and number of lipid droplets. Compared with that in the portal vein region, the number of lipid droplets in the central vein region was greater, but the average probability density was generally lower. Lipid droplets in the central vein region, ranging from large to small, were distributed with an overall spread surrounding the central vein region. In the portal vein region, the number of lipid droplets was small, but the average probability density was large, indicating that the lipid droplets were distributed in a biased manner toward the portal vein.

Based on the probability distribution of lipid droplets at each rank, the entropy was relatively low for Ranks 6 and 5 (including connected clusters), where the lipid droplets were large, indicating that large lipid droplets were aligned and packed densely (Table 3). The entropy tended to be high for the distributions of Ranks 4 and 3, which detected medium and small lipid droplets. This also indicates that the distribution of medium-to-small lipid droplets was more widespread and exhibited an irregular and disordered arrangement.

This study was designed to be generalizable even with a small number of images; however, the small sample size may have affected the generalizability of these results. Therefore, we used the KDE plots and entropy analysis to improve generalizability.

The number of lipid droplets in the histological regions, the central and portal vein regions, were comparable across all ranks. The difference in relative error was small between Ranks 5 and 4, suggesting minimal differences between anatomical regions. Meanwhile, the error was slightly larger for Ranks 6 and 3, suggesting different patterns between anatomical regions depending on rank.

In addition, for Rank 5, the relative error showed a stronger tendency than the other ranks, when using the average probability density. The dataset may have included large variability or outliers. No significant differences were observed for the other ranks (Ranks 6, 4, and 3). No clear differences were observed between anatomical regions. Despite the limited number of images, the average entropy for Rank 5 indicates differences among anatomical regions. Meanwhile, Rank 4 revealed no differences, suggesting comparable entropy between anatomical regions. No significant differences were observed between anatomical regions in other ranks.

In this study, we improved the dynamic management of agent behavior to assess the overall trend of lipid droplet distribution. This facilitated reinforcement learning and increased the search for combination patterns at each rank level. Consequently, the ability to capture the distribution of lipid droplets of similar size and symmetry improved, thereby enhancing practical efficiency (Experiment 3). Although the revised method used a different dataset compared with the conventional method, it achieved an F1 score within the acceptable range (0.9±0.05). The precision and recall were preserved regardless of the F1 score (0.85), which is sufficient in practice (Table 4). Nevertheless, the revised method performed similarly to the conventional method in terms of sensitivity, specificity, accuracy, precision, recall, F1 score, and CV. The main purpose of this study was to train a model with a small dataset. As it is difficult to improve model accuracy by expanding a dataset alone, future research should focus on different algorithms and techniques other than random forests.

The results were similar to those obtained by two pathologists with over 10 years of diagnostic experience. Therefore, both methods enabled the quantitative evaluation of lipid droplet distribution in the central and portal vein regions based

**Table 4. Performance evaluation of the conventional and revised LiDRL methods.**

| Image | Sensitivity | | Specificity | | Accuracy | | Precision | | Recall | | F1-score | | CV | |
|---|---|---|---|---|---|---|---|---|---|---|---|---|---|---|
| | Con | Rev | Con | Rev | Con | Rev | Con | Rev | Con | Rev | Con | Rev | Con | Rev |
| Image001 | 0.95 | 0.91 | 0.84 | 0.72 | 0.895 | 0.815 | 0.861 | 0.765 | 0.95 | 0.91 | 0.902 | 0.831 | 0.891 | 0.914 |
| Image002 | 0.91 | 0.94 | 0.88 | 0.86 | 0.895 | 0.9 | 0.886 | 0.872 | 0.91 | 0.94 | 0.896 | 0.904 | 0.899 | 0.893 |
| Image003 | 0.9 | 0.89 | 0.82 | 0.87 | 0.86 | 0.88 | 0.834 | 0.874 | 0.9 | 0.89 | 0.865 | 0.881 | 0.907 | 0.899 |
| Image004 | 0.87 | 0.89 | 0.85 | 0.85 | 0.86 | 0.87 | 0.854 | 0.857 | 0.87 | 0.89 | 0.861 | 0.872 | 0.905 | 0.896 |
| Image005 | 0.92 | 0.93 | 0.84 | 0.86 | 0.88 | 0.895 | 0.857 | 0.874 | 0.92 | 0.93 | 0.884 | 0.899 | 0.898 | 0.899 |
| Image006 | 0.87 | 0.87 | 0.9 | 0.87 | 0.885 | 0.87 | 0.898 | 0.873 | 0.97 | 0.87 | 0.881 | 0.869 | 0.897 | 0.899 |

*Con; Conventional LiDRL method, Rev; Revised LiDRL method, CV; Cross-validation F1-score.

**Table 5. Comparison of correct answer rates between conventional and revised methods.**

| | Image A | | Image B | |
|---|---|---|---|---|
| Hierarchy | Con | Rev | Con | Rev |
| Rank 6 | Not detected | Not detected | 128.6% (27/21) | 119.0% (25/21) |
| Rank 5 | Not detected | 100% (5/5) | 105.0% (21/20) | 104.8% (22/21) |
| Rank 4 | 107.8% (166/154) | 108.3% (156/144) | 115.7% (287/248) | 115.2% (281/244) |
| Rank 3 | 111.8% (503/450) | 112.6% (492/437) | 113.2% (318/281) | 113.7% (307/270) |

*Con; Conventional LiDRL method, Rev; Revised LiDRL method.

*The numbers in parentheses are the number of detected/correct lipid droplets.

on the average probability density and entropy values of the lipid droplets obtained through KDE. This analysis clarified lipid droplet distribution patterns, suggesting their potential as biomarkers for liver disease.

Using two images from the previous study [21], we compared the detection capabilities of the conventional and revised methods (Table 5). Unlike the revised method, which classified the droplets by size, the conventional method failed to classify medium-sized droplets into Rank 5. The percentage of correctly identified lipid droplets, as assessed by two pathologists, showed little difference between the two methods. Like the conventional method, the revised method occasionally misclassified sinusoids as small lipid droplets. A detected-to-correct lipid droplet ratio greater than 1 indicates over-detection. Therefore, we used the F-score for evaluation. The correct number of lipid droplets is the same number of lipid droplets identified independently by both pathologists. Future studies should aim to improve the accuracy of discriminative learning. In addition, as the difference between the portal and central vein regions may be associated with the underlying diseases, we intend to pursue a more extensive study using comparisons between ranks and entropy analysis.

Lipid droplet analysis is not currently used in the histopathological diagnosis of fatty liver disease; conventionally, qualitative evaluation is performed to confirm the presence or absence and amount of lipid droplets. However, the average probability density and entropy methods used in this study enable the quantification of droplet characteristics such as symmetry and distribution. In future studies, we aim to evaluate the correlations between droplet characteristics and known liver diseases such as fatty liver, fibrosis, and non-alcoholic steatohepatitis.

## Conclusion

The revised LiDRL method facilitates complex modifications to the interaction between the agent and environment by imposing a penalty on the agent's behavior and making certain penalty areas movable. This method can stably and hierarchically organize lipid droplets according to images and capture them quantitatively. It visually captures the number and distribution of lipid droplets using the average probability density and entropy obtained from KDE. It can extract

black-boxed discriminant features from datasets extracted using LiDRL in a manner that aligns with the decision criteria of specialists. In the future, we plan to investigate the characteristics of lipid droplet distribution and explore potential applications for disease classification and diagnosis using machine learning.

## Author contributions

**Conceptualization:** Yoshitomi Harada, Haruto Nishida, Keiko Matsuura.

**Data curation:** Yoshitomi Harada, Haruto Nishida, Keiko Matsuura.

**Formal analysis:** Yoshitomi Harada, Haruto Nishida, Keiko Matsuura.

**Investigation:** Yoshitomi Harada, Haruto Nishida, Keiko Matsuura.

**Methodology:** Yoshitomi Harada, Haruto Nishida, Keiko Matsuura.

**Project administration:** Haruto Nishida, Keiko Matsuura.

**Resources:** Haruto Nishida.

**Software:** Yoshitomi Harada.

**Supervision:** Haruto Nishida, Keiko Matsuura.

**Validation:** Yoshitomi Harada, Haruto Nishida, Keiko Matsuura.

**Visualization:** Yoshitomi Harada.

**Writing – original draft:** Yoshitomi Harada.

**Writing – review & editing:** Haruto Nishida, Keiko Matsuura.

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
