## [Decision Letter · Decision Letter 0]

16 May 2025

PONE-D-25-15072Lipid droplet distribution quantification method based on lipid droplet detection by constrained reinforcement learning

PLOS ONE

Dear Dr. Nishida,

Thank you for submitting your manuscript to PLOS ONE. After careful consideration, we feel that it has merit but does not fully meet PLOS ONE’s publication criteria as it currently stands. Therefore, we invite you to submit a revised version of the manuscript that addresses the points raised during the review process.

Editor Comments

1. Lack of Quantitative Validation of the Detection Algorithm

The manuscript does not provide objective performance metrics for the revised LiDRL method. Although expert agreement is mentioned, no numerical accuracy metrics are reported.

Annotate a subset of your image dataset with expert-labeled ground truth and evaluate model performance using standard detection metrics (e.g., sensitivity, specificity, precision, recall, F1 score). Include a description of how these metrics were calculated, and if possible, inter-rater reliability between experts.

2. Insufficient Sample Size and Lack of Dataset Description

Only six images are used, but there is no information on whether these represent different patients or anatomical zones.

Clearly describe the dataset: how many patients it includes, whether different liver regions were imaged, and how these images were selected. Discuss how the small sample size limits the generalizability of your conclusions.

3. Non-Compliant Data and Code Availability

Your current Data Availability Statement states that data will be made available after acceptance. This is not compliant with PLOS ONE’s policy, which requires all relevant data and code to be available upon submission.

Deposit the source code (for the revised LiDRL algorithm, KDE, and entropy tools) and a representative image dataset or simulated images in a public repository (e.g., GitHub, Zenodo, Figshare). Update your Data Availability Statement to include links or DOIs.

Failure to provide access to code and data prevents other researchers from verifying or building upon your work.

4. Missing Statistical Comparisons and Uncertainty Measures

Your results present averages for entropy and droplet counts, but no statistical tests or confidence intervals are included.

Conduct and report appropriate statistical comparisons (e.g., ANOVA, Mann–Whitney U test) to determine whether differences across ranks or anatomical zones are significant. Include p-values, standard errors, or confidence intervals.

5. Incomplete Biological Interpretation of Findings

The manuscript does not clearly explain what clinical or pathological insights are gained from the lipid droplet distribution patterns you identify.

Expand the Discussion to explain how differences in droplet patterns (entropy, KDE) relate to known liver disease processes, such as steatosis, fibrosis, or NASH. Discuss how your method could assist in diagnosis or research.

6. Language Clarity and Technical Terminology

Some terminology is inconsistently used, and grammatical issues affect clarity. Terms like “rank,” “penalty area,” and “agent” should be defined and used consistently.

Please revise the manuscript for grammar and clarity, ideally using professional language editing services. Ensure all technical terms are defined when first introduced.

7. Missing Ethics Statement in the Main Text

The ethics statement appears only in metadata, not in the Methods section.

Add the following to your Methods section:

“Experiment images were acquired in accordance with ethical research standards (Oita University Faculty of Medicine Ethics approval no. 2568/15 June 2023). The Medical Ethics Review Board approved a waiver of informed consent for this retrospective study.”

8. Figure Quality and Descriptions

Figures were not accessible during editorial review, and current captions are not fully descriptive.

Upload all figures as high-resolution files (TIFF/PNG). Revise figure legends to ensure they describe the image content fully, including any color scales, segmentation overlays, and reference annotations.

We look forward to receiving your revised manuscript.

Kind regards,

Ahmed Abu Siniyeh

Academic Editor

PLOS ONE

Journal Requirements:

Reviewers' comments:

Reviewer's Responses to Questions

**Comments to the Author**

1. Is the manuscript technically sound, and do the data support the conclusions?

Reviewer #1: Yes

Reviewer #2: Partly

2. Has the statistical analysis been performed appropriately and rigorously? 

Reviewer #1: No

Reviewer #2: Yes

3. Have the authors made all data underlying the findings in their manuscript fully available?

Reviewer #1: No

Reviewer #2: No

4. Is the manuscript presented in an intelligible fashion and written in standard English?

Reviewer #1: Yes

Reviewer #2: Yes

5. Review Comments to the Author

Reviewer #1: The paper introduces an improved version of the LiDRL framework, a reinforcement learning-based method designed for detecting lipid droplets in liver pathology images. The proposed enhancements to the agent and environment aim to increase detection accuracy, robustness, and interpretability. The method employs entropy and kernel density estimation to analyze the spatial distribution of lipid droplets, generating heat maps to assist in liver disease diagnosis. While the application of reinforcement learning to histopathology is innovative, the study lacks comparative benchmarks and standardized evaluation metrics. The following suggestions would significantly improve the paper:

1. The paper does not compare the proposed method with established supervised learning models, such as CNNs or U-Net. Including such comparisons would provide a clearer understanding of the relative performance of the proposed method.

2. The paper is missing essential evaluation metrics like precision, recall, F1-score, accuracy, and Intersection over Union (IoU). These metrics should be incorporated to quantify the effectiveness of the detection method more comprehensively.

3. The reinforcement learning framework is under-specified. Critical components, such as the state space, action definitions, and reward structure, need to be more clearly explained to ensure reproducibility and better understanding.

4. The dataset used in the study is not described in sufficient detail. Information regarding the number of samples, the diversity of the dataset, and the annotation process should be provided for greater transparency and to ensure reproducibility.

5. No external validation or cross-dataset testing is conducted, which limits the evidence for the method’s generalizability. Incorporating external validation or testing on different datasets would enhance the credibility of the findings.

6. An ablation study would help identify the individual impact of each modification in the LiDRL framework. This would allow for a clearer understanding of the contribution of each component to the overall performance.

7. The clinical utility of entropy-based heat maps should be better explained, particularly in terms of how they support or correlate with disease diagnosis. This would enhance the clinical relevance of the method.

8. The manuscript would benefit from language and grammar refinement to improve clarity and ensure a more professional presentation.

9. To enhance the reproducibility and transparency of the study, the authors should consider sharing the code, trained models, or annotated datasets. This would allow others to verify the results and contribute to future work in this area.

Reviewer #2: The authors report an enhanced version of the LiDRL method, a reinforcement learning-based approach for the automatic detection of lipid droplets in pathological liver tissue images. By improving both background environmental and agent-side functions, the revised system achieved greater stability and robustness, allowing for consistent extraction and analysis of lipid droplet distribution. The described advancements enable quantification and heat map visualization of droplet patterns, offering potential additional diagnostic indicators for liver disease. While the described method potentially provides a path towards future disease classification and diagnosis using machine learning, no correlation between the achieved quantification of lipid droplet size and distribution in the tested images and their disease state is attempted in the manuscript.

Primary points of concern:

1. The figures and their corresponding legends do not match consistently, which may lead to confusion and misinterpretation of the data. Please ensure that each figure is accurately labeled and that the legends clearly and correctly describe the associated visual content.

2. In tables 1, 2, and 3, the authors need to clearly specify which image set the presented results correspond to. This clarification is essential for the proper interpretation of the presented data.

3. The manuscript states (page 13, line 256) that “the results of this study were similar to those obtained by two pathologists with over 10 years of diagnostic experience,” yet no quantitative evidence is provided to substantiate this claim. Specifically, the number of lipid droplets identified by the experts in the tested images is not reported, which prevents an objective comparison between the expert assessment and the algorithm’s output. As a consequence, the potential over- or underestimation by the previously reported approach and newly reported method in relation to the expert-derived reference values can not be assess by the inclined reader.

Specifically, in table 1 the actual number of droplets identified by human experts for the same image needs to be provided to allow for an assessment of the method's performance.

4. The rank images corresponding to the sample images shown in Figures 3 and 4 should be provided either in a separate figure within the main manuscript or as supplementary material. Including these rank images is important for evaluating the effectiveness and interpretability of the ranking process applied in the analysis.

5. The image quality in Figures 3 and 4 is suboptimal, making it difficult to assess the visualized results. Specifically, the blue boxes referenced in the figure legends are not visible in any of the images (a–f), despite being indicated as present. Moreover, if both blue boxes and shaded areas in varying shades of blue are overlaid on the same images, this may hinder clear differentiation. To improve clarity, the authors are advised to enhance image resolution and use distinctly different colors to represent separate visual elements.

6. The sentence “Moreover, advanced imaging techniques require significant time and resource investment to analyze many images after much training [5–29]” in the Introduction is associated with an unusually large number of references (25 in total) to support a single, general statement. While these references may be thematically related, attributing such a broad claim to a bulk citation appears excessive. The authors are encouraged either to limit the references to the most relevant two or three, or to expand the discussion in this section while keeping overall manuscript limitations in mind to explicitly justify the inclusion of a larger number of sources.

6. PLOS authors have the option to publish the peer review history of their article (what does this mean?). If published, this will include your full peer review and any attached files.

Reviewer #1: No

Reviewer #2: No

---

## [Author Response · Author response to Decision Letter 1]

25 Jun 2025

1. Lack of Quantitative Validation of the Detection Algorithm

The manuscript does not provide objective performance metrics for the revised LiDRL method. Although expert agreement is mentioned, no numerical accuracy metrics are reported.

Annotate a subset of your image dataset with expert-labeled ground truth and evaluate model performance using standard detection metrics (e.g., sensitivity, specificity, precision, recall, F1 score). Include a description of how these metrics were calculated, and if possible, inter-rater reliability between experts.

Thank you for your comment. In addition to the results of Experiment 3, we have added the following metrics: “Performance evaluation of the conventional method and the proposed method” and “Calculation method,” which are summarized in Table 4. Changes were made, as follows:

“Experiment 3: Comparison between the conventional LiDRL method and the revised LiDRL method

We compared the performance of the conventional and revised LiDRL methods, leading us to modify the experimental and calculation protocols accordingly. The performance of both models was evaluated using sensitivity, specificity, accuracy, precision, recall, F1 score, and CV (Equations 3 to 9) values, obtained over an average of five runs. In addition, as the performance evaluation of the conventional LiDRL method [21] used the F1 score based on 10-fold cross-validation, the same comparison criteria were used for the revised LiDRL method.

Sensitivity= TP / (TP + FN) (3)

Specificity= TN / (TN + FP) (4)

Accuracy = (TP + TN) / (TP + TN + FP + FN) (5)

These measures represent the overall correctness of the classification model.

Precision = TP / (TP + FP) (6)

It evaluates the proportion of predicted positives that are truly positive.

Recall = TP / (TP + FN) (7)

It indicates the ability of the model to accurately capture true positives.

F1 score = 2 × Precision × Recall / (Precision + Recall) (8)

It balances precision and recall, which is useful when classes are imbalanced.

Cross-validation F1-score = 1/k ∑_(i = 1)^k▒〖F1〗_i (k = 10) (9)

TP (True Positive): Correctly predicted positives

TN (True Negative): Correctly predicted negatives

FP (False Positive): Incorrectly predicted positives

FN (False Negative): Incorrectly predicted negatives

Furthermore, to compare and evaluate the performance of the revised method versus the conventional method, two pathologists identified lipid droplets on the same images.”

Table 4. Performance evaluation of the Conventional LiDRL method and the Revised LiDRL method.

Image Sensitivity Specificity Accuracy Precision Recall F1-score CV

Con Rev Con Rev Con Rev Con Rev Con Rev Con Rev Con Rev

Image001 0.95 0.91 0.84 0.72 0.895 0.815 0.861 0.765 0.95 0.91 0.902 0.831 0.891 0.914

Image002 0.91 0.94 0.88 0.86 0.895 0.9 0.886 0.872 0.91 0.94 0.896 0.904 0.899 0.893

Image003 0.9 0.89 0.82 0.87 0.86 0.88 0.834 0.874 0.9 0.89 0.865 0.881 0.907 0.899

Image004 0.87 0.89 0.85 0.85 0.86 0.87 0.854 0.857 0.87 0.89 0.861 0.872 0.905 0.896

Image005 0.92 0.93 0.84 0.86 0.88 0.895 0.857 0.874 0.92 0.93 0.884 0.899 0.898 0.899

Image006 0.87 0.87 0.9 0.87 0.885 0.87 0.898 0.873 0.97 0.87 0.881 0.869 0.897 0.899

*Con; Conventional LiDRL method, Rev; Revised LiDRL method, CV; Cross-validation F1-score

We have also revised the discussion section to reflect these changes. The discussion was revised as follows:

“ In this study, we improved the dynamic management of the behavior of the agent and obstacle in order to assess the overall trend of lipid droplet distribution. This facilitated reinforcement learning and increased the search for combination patterns at each rank level. Consequently, it improved the ability to capture the distribution of lipid droplets of similar size and symmetry, thereby improving practical efficiency (Experiment 3). Although the revised method used a dataset different from that used by the conventional method, it achieved an F1 score within the acceptable range (0.9 ± 0.05). The precision and recall were preserved regardless of the F1 score of 0.85, which suffices in practice (Table 4). Nevertheless, the revised method demonstrated similar performance to the conventional method in terms of sensitivity, specificity, accuracy, precision, recall, F1 score, and CV. Although the accuracy of the model may be enhanced in future studies through the utilization of larger training datasets, the primary objective of our study was to train the model on a small dataset.”

In this study, we did not compare our method with existing supervised learning models such as CNN and U-Net. These models involve image classification inaccuracy and overfitting risk, and make it difficult to preserve segmentation. To clarify this point, have reorganized the references [5-20] and added the following information to the introduction:

“To address this, several methods using neural networks have been proposed [5-14]. Advanced techniques, such as Convolutional Neural Network are suitable for image classification but have limitations for preserving fine structures such as segmentation [15-17]. A method using U-net has been proposed; however, it involves a risk of increased computational complexity, memory requirements, and overfitting due to the symmetric structure of the encoder and decoder [18-20]. These methods were deemed unsuitable for this study and therefore were not utilized.”

In addition, to compare the present and conventional method, the numbers of lipid droplets identified by two pathologists on the same images are presented in Table 5.

Table 5. Comparison of correct answer rates between conventional and revised methods.

Image A Image B

Hierarchy Con Rev Con Rev

Rank6 Not detected Not detected 77.8% (21/27) 84.0% (21/25)

Rank5 Not detected 100% (5/5) 95.2% (20/21) 95.5% (21/22)

Rank4 92.8% (154/166) 92.3% (144/156) 86.4% (248/287) 86.8% (244/281)

Rank3 89.5% (450/503) 88.8% (437/492) 88.4% (281/318) 87.9% (270/307)

*Con; Conventional LiDRL method, Rev; Revised LiDRL method

*The numbers in parentheses are the number of correct /detections lipid droplets

We revised the discussion section to reflect these changes:

“Using two images from the same study [21], we compared the detection capabilities of the conventional and revised methods (Table 5). First, in contrast to the revised method, which classified the droplets by size, the conventional method failed to classify medium-sized lipid droplets into Rank 5. The percentage of correct answers obtained from the evaluation of the detected lipid droplets by the two pathologists showed little difference between the two methods. Similar to the conventional method, the revised method occasionally misclassified sinusoids as lipid droplets when assessing them as potential small droplets. Future studies should aim to improve the accuracy of discriminative learning.”

2. Insufficient Sample Size and Lack of Dataset Description

Only six images are used, but there is no information on whether these represent different patients or anatomical zones.

Clearly describe the dataset: how many patients it includes, whether different liver regions were imaged, and how these images were selected. Discuss how the small sample size limits the generalizability of your conclusions.

Thank you for your comment. We have added the following information to the methods section:

“In this study, we used six images, with two images taken from each of three different patients. These included the central and portal veins (each image size: 1600 × 1200 pixels, grayscale: color, 8 bits). The images were randomly selected based on anatomical requirements (one portal vein region, one central vein region). However, the results of an analysis using a limited number of images may have limited applicability when extrapolated to the entire patient population. Therefore, we estimated the shape of the distribution using KDE plots and quantified the randomness and information content of the dataset using entropy analysis. Using KDE, we effectively estimated the shape and pattern of data distribution even with a limited number of images, thus facilitating more accurate feature extraction. By evaluating the complexity of the information (data diversity and bias) using entropy analysis, we obtained the information required for suitable model selection in the next analysis.”

We have added the following to the discussion section:

“This study was designed to be generalizable even with a small number of images; however, the small sample size may have affected the generalizability of these results. Therefore, we used the KDE plots and entropy analysis to improve generalizability.”

3. Non-Compliant Data and Code Availability

Your current Data Availability Statement states that data will be made available after acceptance. This is not compliant with PLOS ONE’s policy, which requires all relevant data and code to be available upon submission.

Deposit the source code (for the revised LiDRL algorithm, KDE, and entropy tools) and a representative image dataset or simulated images in a public repository (e.g., GitHub, Zenodo, Figshare). Update your Data Availability Statement to include links or DOIs.

Failure to provide access to code and data prevents other researchers from verifying or building upon your work.

Thank you for your comment. We have made the required information available as "Revised LiDRL Algorithm, KDE, and Entropy Analysis" using Google Collaboratory.

https://zenodo.org/records/15733729

4. Missing Statistical Comparisons and Uncertainty Measures

Your results present averages for entropy and droplet counts, but no statistical tests or confidence intervals are included.

Conduct and report appropriate statistical comparisons (e.g., ANOVA, Mann–Whitney U test) to determine whether differences across ranks or anatomical zones are significant. Include p-values, standard errors, or confidence intervals.

Thank you for your comment. We have added the following information in the methods section (Experiment 2):

“The average values obtained from the six images were evaluated using the Mann–Whitney U test, p-value, and standard error to determine whether there were significant differences between the anatomical regions.”

Table 3 was added to the results section:

Table 3. Variations between the central vein region and the portal vein region [six images]

Category Hierarchy Central vein region Portal vein region Mann-Whitney U p-value Relative Error [%]

Lipid Droplets Count Rank 6 36.0 23.3 7.0000 0.4000 35.19

Rank 5 55.7 12.7 6.0000 0.5050 6.59

Rank 4 241.0 256.0 4.0000 1.0000 6.22

Rank 3 283.3 220.3 8.0000 0.2000 22.24

Average Probability Density Rank 6 0.0186 0.0296 2.0000 0.4000 59.29

Rank 5 0.0115 0.0637 0.0000 0.0765 452.13

Rank 4 0.0027 0.0026 5.0000 1.0000 3.80

Rank 3 0.0023 0.0030 1.0000 0.2000 33.57

Average Entropy Rank 6 2.557 2.268 7.0000 0.4000 11.29

Rank 5 2.844 1.831 9.0000 0.0765 35.62

Rank 4 3.777 3.808 4.0000 1.0000 0.84

Rank 3 3.881 3.709 8.0000 0.2000 4.44

We have also added information to the discussion section, as follows:

“The number of lipid droplets in the histological regions, the central and portal vein regions, were comparable across all ranks. Regarding the relative error, the difference was small for Rank 5 and Rank 4, suggesting that the differences between anatomical regions were relatively small. Meanwhile, the error was slightly larger for Rank 6 and Rank 3, suggesting that the patterns between anatomical regions may differ depending on the rank.

In addition, the relative error for rank 5 when using the average probability density showed a stronger tendency than the other ranks. The dataset may have included large variability or outliers. No significant differences were observed for the other ranks (Ranks 6, 4, and 3). No clear differences were observed between anatomical regions. Despite a limited number of images used, the average entropy for Rank 5 indicates differences among anatomical regions. Meanwhile, Rank 4 revealed no differences, suggesting comparable entropy between anatomical regions. There were no significant differences between anatomical regions in other ranks.”

We have added the following to the discussion section:

“In addition, as the difference between the portal and central vein regions may be associated with the underlying diseases, our objective is to conduct a more extensive study in the future, using comparisons between ranks, and entropy analysis.”

5. Incomplete Biological Interpretation of Findings

The manuscript does not clearly explain what clinical or pathological insights are gained from the lipid droplet distribution patterns you identify.

Expand the Discussion to explain how differences in droplet patterns (entropy, KDE) relate to known liver disease processes, such as steatosis, fibrosis, or NASH. Discuss how your method could assist in diagnosis or research.

We have added the following to the discussion section:

“Lipid droplet analysis is not currently used in the histopathological diagnosis of fatty liver disease; conventionally, qualitative evaluation was mainly performed to confirm the presence or absence and amount of lipid droplets. However, the average probability density and entropy methods used in this study enables the quantification of droplet characteristics such as symmetry and distribution. In our future studies, we aim to evaluate the correlation between droplet characteristics and known liver diseases such as fatty liver, fibrosis, and NASH.”

6. Language Clarity and Technical Terminology

Some terminology is inconsistently used, and grammatical issues affect clarity. Terms like “rank,” “penalty area,” and “agent” should be defined and used consistently.

Please revise the manuscript for grammar and clarity, ideally using professional language editing services. Ensure all technical terms are defined when first introduced.

The manuscript was edited by a professional medical editing service.

We have included the relevant definitions in Fig. 1.

“We used the following terms to explain our algorithm: filter size, a square frame with one side as large as the maximum diameter that can contain only one lipid droplet; rank, order within a 3 × 3 filter region in a binary image; environment, a 4 × 10 grid; reward, the ratio of overlapping region to detected lipid droplet region (Fig 1).

Fig 1. Definition of reward: Overlapping region of the newly extracted region ([21]).

agent, a subject that moves, learns, and selects actions that maximize rewards based on the state of the environment; state, the position where the agent is in the environment. It was defined as a filter size and rank; action, the movement the agent can choose in each state. The agent can move leftward or downward; Q value, the total reward expected from a specific action in a certain state; penalty area, if the agent enters this area, a negative reward is imposed, and the agent is forced to return to the starting point. This area maintains accuracy by repeating a certain number of movements; start, the initial position of the agent; goal, the final position the agent reaches.”

7. Missing Ethics Statement in the Main Text

The ethics statement appears only in metadata, not in the Methods section.

Add the following to your Methods section:

“Experiment images were acquired in accordance with ethical research standards (Oita University Faculty of Medicine Ethics approval no. 2568/15 June 2023). The Medical Ethics Review Board approved a waiver of informed consent for this retrospective study.”

Thank you for your comment. We have included an ethics statement in the manuscript.

8. Figure Quality and Descriptions

Figures were not accessible during editorial review, and current captions are not fully descriptive.

Upload all figures as high-resolution files (TIFF/PNG). Revise figure legends to ensure they describe the image content fully, including any color scales, segmentation overlays, and reference annotations.

Thank you for your comment. We have made these changes, as required.

---

## [Decision Letter · Decision Letter 1]

21 Jul 2025

PONE-D-25-15072R1Lipid droplet distribution quantification method based on lipid droplet detection by constrained reinforcement learningPLOS ONE

Dear Dr. Nishida,

Thank you for submitting your manuscript to PLOS ONE. After careful consideration, we feel that it has merit but does not fully meet PLOS ONE’s publication criteria as it currently stands. Therefore, we invite you to submit a revised version of the manuscript that addresses the points raised during the review process.

We look forward to receiving your revised manuscript.

Kind regards,

Ahmed Abu Siniyeh

Academic Editor

PLOS ONE

Journal Requirements:

Reviewers' comments:

Reviewer's Responses to Questions

**Comments to the Author**

1. If the authors have adequately addressed your comments raised in a previous round of review and you feel that this manuscript is now acceptable for publication, you may indicate that here to bypass the “Comments to the Author” section, enter your conflict of interest statement in the “Confidential to Editor” section, and submit your "Accept" recommendation.

Reviewer #1: All comments have been addressed

Reviewer #2: (No Response)

2. Is the manuscript technically sound, and do the data support the conclusions?

Reviewer #1: Yes

Reviewer #2: Partly

3. Has the statistical analysis been performed appropriately and rigorously? 

Reviewer #1: Yes

Reviewer #2: Yes

4. Have the authors made all data underlying the findings in their manuscript fully available?

Reviewer #1: Yes

Reviewer #2: Yes

5. Is the manuscript presented in an intelligible fashion and written in standard English?

Reviewer #1: Yes

Reviewer #2: Yes

6. Review Comments to the Author

Reviewer #1: I am pleased to recommend acceptance of this manuscript. However, I suggest that the authors carefully correct minor typographical errors and ensure that all plots and figures are properly arranged before final submission.

Reviewer #2: The authors have clearly taken the comments into account and provided a significantly improved version of the manuscript. The structure and clarity of the revised text have improved, and several earlier concerns have been adequately addressed.

However, the low number of training and validation samples remains a limitation of this study. While it would have been preferable to expand the number of images in both the training and especially the validation dataset, the authors now acknowledge and discuss the limitations arising from this issue in the revised manuscript. This is appreciated and can be regarded as having been adequately addressed.

A minor correction is recommended in the legend of the revised Table 5. The phrase "number of correct /detections lipid droplets" likely intends to mean "correct/detected lipid droplets". More importantly, expressing this ratio as detected/correct would be clearer, especially since the method appears to over-detect lipid droplets.

Lastly, in response to reviewer comment 1 and the updated Table 5, the authors note that two pathologists’ annotations were used for comparison. It would be helpful if the authors could also state the total number of lipid droplets identified by the two pathologists (or their average, if the counts differ) on the validation images independent of the hierarchic rank of the employed analysis method. This would give readers a better understanding of the clinical relevance of the proposed method, independent of the ranking process.

7. PLOS authors have the option to publish the peer review history of their article (what does this mean?). If published, this will include your full peer review and any attached files.

Reviewer #1: No

Reviewer #2: No

---

## [Author Response · Author response to Decision Letter 2]

12 Aug 2025

Reviewer #1: I suggest that the authors carefully correct minor typographical errors and ensure that all plots and figures are properly arranged before final submission.

Thank you for your comments. We have proofread the whole document again and made the necessary corrections.

Reviewer #2: The low number of training and validation samples remains a limitation of this study. While it would have been preferable to expand the number of images in both the training and especially the validation dataset, the authors now acknowledge and discuss the limitations arising from this issue in the revised manuscript. This is appreciated and can be regarded as having been adequately addressed.

Thank you for your constructive comments. As you pointed out, we also recognize the small number of cases as a limitation. The following comments have been added and corrected in the discussion on page 17: “The main purpose of this study was to train a model with a small dataset. Given the difficulty in improving accuracy through dataset expansion, future research should explore alternative algorithms beyond random forests.” It remains a challenge, and we plan to continue our research on this topic in the future.

A minor correction is recommended in the legend of the revised Table 5. The phrase "number of correct /detections lipid droplets" likely intends to mean "correct/detected lipid droplets". More importantly, expressing this ratio as detected/correct would be clearer, especially since the method appears to over-detect lipid droplets.

The legend in Table 5 has been revised to read, "The numbers in parentheses are the number of detected/correct lipid droplets." The numbers in Table 5 have also been changed accordingly.

Lastly, in response to reviewer comment 1 and the updated Table 5, the authors note that two pathologists’ annotations were used for comparison. It would be helpful if the authors could also state the total number of lipid droplets identified by the two pathologists (or their average, if the counts differ) on the validation images independent of the hierarchic rank of the employed analysis method. This would give readers a better understanding of the clinical relevance of the proposed method, independent of the ranking process.

The number of lipid droplets that two doctors simultaneously identified is shown on the verification image. Additionally, the following comment has been added to the discussion on p.18: "A detected-to-correct lipid droplet ratio greater than 1 indicates over-detection. Therefore, we used the F-score for evaluation. The correct number of lipid droplets is the same number of lipid droplets identified independently by both pathologists."

---

## [Decision Letter · Decision Letter 2]

7 Sep 2025

Lipid droplet distribution quantification method based on lipid droplet detection by constrained reinforcement learning

PONE-D-25-15072R2

Dear Dr. Nishida,

We’re pleased to inform you that your manuscript has been judged scientifically suitable for publication and will be formally accepted for publication once it meets all outstanding technical requirements.

Kind regards,

Ahmed Abu Siniyeh

Academic Editor

PLOS ONE

Additional Editor Comments (optional):

Reviewer #2:

Reviewers' comments:

Reviewer's Responses to Questions

**Comments to the Author**

1. If the authors have adequately addressed your comments raised in a previous round of review and you feel that this manuscript is now acceptable for publication, you may indicate that here to bypass the “Comments to the Author” section, enter your conflict of interest statement in the “Confidential to Editor” section, and submit your "Accept" recommendation.

Reviewer #2: All comments have been addressed

2. Is the manuscript technically sound, and do the data support the conclusions?

Reviewer #2: Yes

3. Has the statistical analysis been performed appropriately and rigorously? 

Reviewer #2: Yes

4. Have the authors made all data underlying the findings in their manuscript fully available?

Reviewer #2: Yes

5. Is the manuscript presented in an intelligible fashion and written in standard English?

Reviewer #2: Yes

6. Review Comments to the Author

Reviewer #2: In the latest version of the revised manuscript, all previously raised points have been thoroughly reviewed and addressed. In light of the satisfactory revisions, the reviewer has no substantive objections to the manuscript being accepted for publication in its current form.

7. PLOS authors have the option to publish the peer review history of their article (what does this mean?). If published, this will include your full peer review and any attached files.

Reviewer #2: No

---

## [Editor Report · Acceptance letter]

PONE-D-25-15072R2

PLOS ONE

Dear Dr. Nishida,

I'm pleased to inform you that your manuscript has been deemed suitable for publication in PLOS ONE. Congratulations! Your manuscript is now being handed over to our production team.

Kind regards,

on behalf of

Dr. Ahmed Abu Siniyeh

Academic Editor

PLOS ONE